# In-season training responses and perceived wellbeing and recovery status in professional soccer players

Nuno Mateus[1,2]☯*, Bruno Gonçalves[3,4,5]☯, Jose Luis Felipe[6]☯, Javier Sánchez-Sánchez[6]☯, Jorge Garcia-Unanue[7]☯, Anthony Weldon[8]☯, Jaime Sampaio[1,2]☯

1 Research Center in Sports Sciences, Health Sciences and Human Development, CIDESD, CreativeLab Research Community, Vila Real, Portugal, 2 Department of Sports Science, Exercise and Health, University of Trás-os-Montes e Alto Douro, Vila Real, Portugal, 3 Departamento de Desporto e Saúde, Escola de Saúde e Desenvolvimento Humano, Universidade de Évora, Évora, Portugal, 4 Comprehensive Health Research Centre (CHRC), Universidade de Évora, Évora, Portugal, 5 Portugal Football School, Portuguese Football Federation, Oeiras, Portugal, 6 Universidad Europea de Madrid, Faculty of Sport Sciences, Villaviciosa de Odón, Madrid, Spain, 7 Universidad de Castilla-La Mancha, IGOID Research Group, Toledo, Spain, 8 Technological and Higher Education Institute of Hong Kong (THEi), Chai Wan, Hong Kong

☯ These authors contributed equally to this work.
* nuno_mateus23@hotmail.com

**Data Availability Statement:** The data is only available on request, to protect the subjects' confidentiality and privacy. Interested researchers

## Abstract

This study aimed to describe professional soccer players' training responses during a competitive season and to investigate the relationship between these responses with wellbeing and recovery indices. Thirteen professional soccer players from the same Spanish Second Division team were monitored during a sixteen-week in-season period. Players' external loads were analyzed using global positioning measurement units (GPS). Additionally, subjective reporting of sleep quality, sleep duration, fatigue, muscle soreness, and stress were assessed with a customized wellness questionnaire at the beginning of each training session. A two-step cluster analysis identified profiles of different training responses generally described as lower-demand sessions, intermediate-demand sessions, running-based sessions, and sprint-based sessions; which were discriminated by different total distance covered and high-intensity actions. Interestingly, no probabilistic interactions were found between these training responses with wellbeing and recovery markers (i.e., Bayes factor < 1 suggesting no evidence, for all the variables). These findings may raise concerns about using self-reporting tools, as they show that players' wellness data is probably not accounted for when coaching staff plan and optimize the training process. However, results should be interpreted with caution, due to representing a single team and coaching staff.

## Introduction

Soccer is a complex team sport characterized by unpredictable activity patterns during the match, as players frequently change from short multidirectional high-intensity displacements with longer periods of low-intensity activity [1, 2]. Consequently, coaches frequently aim to

may contact the board from the Research Center in Sports Sciences, Health Sciences and Human Development to request access to the data (CIDESD, geral@utad.pt).

**Funding:** This work was supported by the Portuguese Foundation for Science and Technology (FCT), and European Social Fund (ESF), through a Doctoral grant endorsed to the first author (SFRH/BD/138499/2018) under the Human Potential Operating Program (POPH); and also, by the 'Fondo Europeo de Desarrollo Regional, Programa Operativo de la Región de Castilla-La Mancha' endorsed to the fifth author (2018/11744). Furthermore, the funders had no role in study design, data collection and analysis, decision to publish, or preparation of the manuscript.

**Competing interests:** The authors have declared that no competing interests exist.

mimic the intensity and movement patterns of the match by intensifying training session demands, to prepare players to keep high-performance levels throughout a competitive season [3–5]. To provide comprehensive information about players' load, both in training and competition environments, global positioning measurement technology (GPS) is commonly used, allowing the coaching staff to manage and quantify each players' effort, to make informed decision regarding player's performance, while also minimizing injury risk [6–8]. Furthermore, this monitoring approach enables coaches to understand the physical demands of each playing position and the conditioning needs for individual player's within the team [9]. This knowledge also facilitates training' periodization, once the training plan accounts for the competition schedule, team' technical-tactical principles, players' conditioning requirements, and physical recovery status [7, 9–11]. A crucial feature of team sports is the coach's ability to administer appropriate training volumes and intensities that fulfil all these requirements, which leads to a tailored training stimulus throughout the training week, resulting in sessions with different demands and specific characteristics [7, 9, 12]. Although, recent studies have started to describe load variations, according to the match day in more detail [7, 9, 12, 13], the literature generally examines exclusively the total weekly training load volume [14, 15], which conceals meaningful load variations (i.e., high-intensity actions and running volumes) and neglects the benefits that a detailed characterization could provide.

Previous soccer studies have identified that post-training and post-match fatigue results from a combination of glycogen depletion, muscle damage, and mental fatigue [16]. Thus, considering that elite soccer is played throughout a highly congested schedule, optimal recovery strategies are required to decrease fatigue and reduce the injury risk [16]. In this regard, sleep quality and duration are considered important psychological and physiological functions that may substantially contribute to the recovery process in elite soccer players [17, 18]. Sports science literature has documented that sleep disturbance increases the risk, prevalence and severity of musculoskeletal injuries and is associated with cognitive, technical and physical performance impairment; whereas healthier sleep habits may enhance physical and technical performance [17, 18]. Despite this evidence, elite soccer is associated with sleep performance impairment specifically through the time and schedule of training and competition events and long-haul traveling [19]. Concomitantly, an inappropriate balance between training stimulus and recovery along the season is suggested as a catalyst to overreaching and muscle damage in elite soccer players [5, 17, 20]. Subsequently, the fatigue and muscle soreness accumulation may limit motor processing and perception, contributing to unsuccessful technical actions and poor physical performance [13, 21, 22].

In line with this reasoning, currently, the measurement of wellbeing and recovery indices is a common procedure in professional soccer [8, 12, 13, 23, 24]. Over the last decade, researchers have used wellness questionnaires to monitor stress, recovery, and the psychometric status of players, across various sports and levels of competition [13, 23, 24]. In addition to being non-invasive and suitable to collect data over significant periods at home or while traveling, self-reported methods have been described as prominent and effective to detect early signs of tiredness and overtraining, and, eventually, diminish the risk of injury and illness [7, 13]. Nevertheless, despite their benefits, reservations concerning its application have been raised, specifically, that they depend on the players' cognitive focus (i.e., data is susceptible to being manipulated and over/underestimated), are arduous to interpret and non-specific to distinguish individuals within the same sport [23, 24]. Hence, there is no full consensus about how wellness questionnaires should be used to optimize the training process, which is surprising, considering that players' pre-training state can compromise the daily training intention [13].

Therefore, it is important to understand how soccer players' training responses fluctuate during the in-season. We hypothesized that coaches' application of training methods, that

mimic game demands and include preventive strategies (e.g., unload players prior to match days) will result in different training loads. Furthermore, proper sleep and wellbeing may be associated with higher training stimulus, while poor sleep and recovery indices may lead to lower training loads as a coaching strategy to cope with the players' pre-training wellness status. Consequently, the purpose of this study was to describe professional soccer players' training responses during the in-season and to investigate whether these different responses are associated with the players' wellbeing and recovery indices.

## Materials and methods

### Participants

Twenty-two male professional outfield soccer players (age, 26.14 ± 3.89 years old; weight, 67.22 ± 3.44 kg; height, 182 ± 9.2 cm) competing in the Spanish Second Division (level two in Spain professional soccer) volunteered to participate in the study. Criteria for inclusion were applied to ensure players were familiarized with all procedures (customized digital perceived wellness questionnaire and training with GPS units) during the pre-season and have regular participation in most weekly training sessions. Players that performed fewer than ten training sessions suffered prolonged injuries (i.e., > two weeks), or were medicated during the study period were excluded from the analysis [9], which led to a final sample of thirteen players (age, 27.42 ± 4.68 years old; weight, 69.83 ± 4.22 kg; height, 179 ± 6 cm). All players were informed about the rights and commitments of participating in this study and provided informed and written consent before the study commenced. No players reported any musculoskeletal, neurological, or orthopedic injury that could impair their participation. The study protocol was approved and followed the guidelines stated by the local Institution–Ethics Committee of the European University of Madrid (CIPI35/2019)–and in conformity with the recommendations of the Declaration of Helsinki.

### Design

Training load and pre-training wellness indices data for all players were collected over a 16-week in-season period (November-February), without any intervention of the research team regarding training volume and intensity, neither to wellness status perceived by the players (Fig 1). Participants trained on a full-time basis and played competitive fixtures within the Spanish Second Division and Copa del Rey during the 2017–2018 season. Throughout the data-collection period, the team competed in 17 official matches, which often meant that the team played 1 match per week. To increase the reliability of the data collected, and to respect the studies aims, only pitch-based team training sessions (i.e., regular training, including starters and nonstarters) were considered. Furthermore, match data was also not included in this study, as players were monitored with a different tracking system, thus preventing data bias. To eliminate potential effects of training time on the variation of the players' training responses, training sessions shorter than one hour (65 cases), longer than two hours (3 cases),

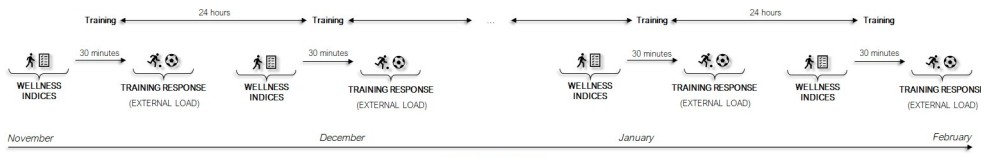

**Fig 1. Representation of data collection period.** The ellipsis represents the sequence of the training sessions throughout the 16-weeks of the in-season period.

and data referring to players who did not undertake the whole training session (e.g., injury, 5.23 ± 3.17 training sessions per player) were not analyzed. Consequently, the final sample gathered 678 individual records.

## Data collection

**Training load monitoring.** The training load was collected using individual Viper GPS units (Viper, STATSports, Newry, Ireland). The validity and reliability of the STATSports Viper system were previously reported, and their operation and handling are documented elsewhere [25]. After each training session, training load data was downloaded using the respective software package (Viper PSA software, STATSports, Newry, Ireland) and exported for analysis [25]. The following variables were selected for analysis: total distance covered, distance covered at different velocities, the number of sprints, high-intensity accelerations (>3 m/s$^2$), high-intensity decelerations (<-3 m/s$^2$), impacts (>8 G's forces), and player load [26]. The distance covered at different movement speeds were adapted from previous soccer studies and standardized into six-speed categories: standing (<0.6 km/h), walking (0.7–7.1 km/h), jogging (7.2–14.3 km/h), running (14.4–19.7 km/h), high-speed running (19.8–25.1 km/h), and sprinting (>25.1 km/h) [1, 2]. To ensure the validity of the metrics used, a small amount of the players' bidimensional coordinates obtained through the Viper units were exported and computed using dedicated codes written in Matlab® (MathWorks, Inc., Massachusetts, USA); afterward, the reliability of the data was verified through a Bland-Altman graph. Furthermore, all variables were normalized according to the time on the pitch during each training session to provide an understanding of session intensity [13, 27].

**Perceived wellness indices.** The players were instructed to complete a customized digital perceived wellness questionnaire (Fig 2), thirty minutes before each training session, at the facilities of the club, which they were familiarized with during the pre-season. The questionnaire was designed to be brief, precise, and based on the components of self-perceived tools used to assess players' wellness in the literature [23, 24]. Each player was asked to provide details about the following wellbeing and recovery variables: sleep quality and duration, fatigue, muscle soreness, and stress. All parameters were measured using a Likert scale ranging from 1 to 10, where 1 indicated "very, very low" (fatigue, stress, and muscle soreness) or "poor" (sleep quality and time), and 10 indicated "very, very high" (fatigue, stress, and muscle soreness) or "optimal" (sleep quality and duration).

## Statistical analysis

A two-step cluster analysis with log-likelihood as the distances measure and Schwartz's Bayesian criterion was carried out to describe the players' responses into different groups according to all training variables. Afterward, a Bayesian ANOVA [28] was used to quantify the predictive influence of the wellbeing and recovery variables on the clustered groups obtained previously. The wellbeing and recovery variables were considered as dependent variables, whereas the training clusters were considered as a fixed factor. The Cauchy prior width was set at *r* scale fixed effects = 0.5 [29]. The two-step cluster analysis was conducted using the Statistical Package for the Social Sciences software (IBM Corp. Released 2019. IBM SPSS Statistics for Windows, Version 26.0. Armonk, NY: IBM Corp.). The Bayesian ANOVA was performed using JASP software (JASP Team 2019. JASP for Windows, Version 0.11.1, computer software).

## Results

The mean and standard deviation from all training-related variables, according to the clusters and their predictor importance for differentiating each cluster are presented in Table 1. As a

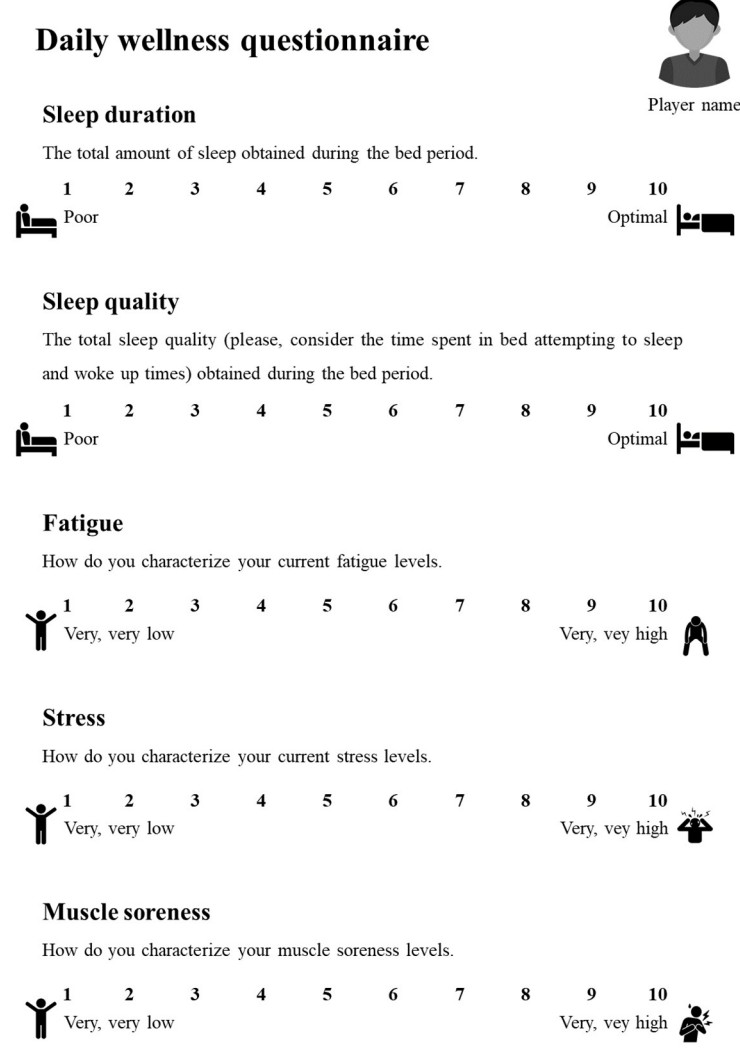

**Fig 2. Daily wellness questionnaire administered to all players before each training session.**

complement, Fig 3 portrays the distribution of the training variables among the clusters obtained. Variables such as total distance covered, sprints, player load, and distance covered at high-speed running and sprinting, exhibited higher importance to discriminate the clusters; conversely, accelerations, decelerations, and distance covered at standing reveal less influence in differentiating the clusters. Most data were grouped in cluster INT, sessions described as *intermediate-demand sessions* (40%), followed by cluster RUN, *running-based sessions* (21.8%), cluster SPR, *sprint-based sessions* (19.9%), and cluster LOW, *lower-demand sessions* (18.3%). Cluster LOW incorporated the lower training loads. Cluster INT gathered significantly higher training loads than the previous cluster, but lower training loads than the following clusters. Cluster RUN presented the highest mean for distance covered (79.2±7.92), distance covered at low speeds thresholds (i.e., standing (36.34±4.86), walking (14.92±3.14), jogging (12.1±3.90) and running (14.23±7.61)), impacts (5.13±2.51) and player load (1.25±0.17). Cluster SPR assembled the highest values for sprints (0.12±0.05), accelerations (0.59±0.23), decelerations

**Table 1. Means, standard deviations and predictor importance from the obtained clusters.**

| Variables | Cluster (mean ± SD) (CV%) | | | | Predictor importance |
|---|---|---|---|---|---|
| | Cluster LOW (n = 124) | Cluster INT (n = 271) | Cluster RUN (n = 148) | Cluster SPR (n = 135) | |
| Distance covered (m) | 50.38±7.55 (14.9) | 68.06±6.05 (8.9) | 79.2±7.92 (10.0) | 74.56±8.67 (11.6) | 1.00 |
| Standing (m) | 30.55±5.45 (17.8) | 35.18±4.24 (12.0) | 36.34±4.86 (13.4) | 35.27±4.56 (12.9) | 0.16 |
| Walking (m) | 8.53±2.02 (23.7) | 12.54±2.19 (17.5) | 14.92±3.14 (21.0) | 13.48±2.81 (20.9) | 0.52 |
| Jogging (m) | 5.29±1.81 (34.2) | 9.51±2.63 (27.7) | 12.1±3.90 (32.3) | 9.49±2.32 (24.5) | 0.47 |
| Running (m) | 5.13±2.58 (50.3) | 9.48±3.01 (31.8) | 14.23±7.61 (53.5) | 11.74±2.97 (25.3) | 0.38 |
| High-speed running (m) | 0.61±0.55 (90.4) | 0.95±0.52 (54.7) | 1.15±0.69 (59.5) | 2.68±0.88 (32.8) | 0.8 |
| Sprinting (m) | 0.27±0.37 (140.2) | 0.39±0.36 (93.4) | 0.45±0.37 (82.8) | 1.91±1.04 (54.5) | 0.77 |
| Sprints (a.u) | 0.02±0.03 (119.3) | 0.03±0.02 (76.7) | 0.04±0.03 (70.9) | 0.12±0.05 (40.9) | 0.83 |
| Accelerations (a.u) | 0.44±0.21 (47.2) | 0.50±0.20 (39.9) | 0.57±0.22 (38.1) | 0.59±0.23 (38.6) | 0.06 |
| Decelerations (a.u) | 0.19±0.12 (65.3) | 0.25±0.20 (80.8) | 0.20±0.12 (62.6) | 0.45±0.30 (67.5) | 0.19 |
| Impacts (a.u) | 1.65±1.23 (74.7) | 2.05±1.16 (56.8) | 5.13±2.51 (49.0) | 2.41±1.42 (58.9) | 0.49 |
| Player Load (a.u) | 0.77±0.15 (20.2) | 0.94±0.15 (15.9) | 1.25±0.17 (13.3) | 1.08±0.21 (19.2) | 0.66 |

(0.45±0.03), and distance covered at high-speed thresholds (i.e., high-speed running (2.68 ±0.88), and sprinting (1.91±1.04)).

The inferences of the Bayesian ANOVA are shown in Tables 2 and 3, and Fig 4. The Bayes factor suggested an absence of a relationship between the training clusters and both the sleep duration and the fatigue levels predicted by the players (sleep duration: $BF_{10} = 0.027$, $BF_{01} = 1/0.027 = 37.037$; fatigue: $BF_{10} = 0.028$, $BF_{01} = 1/0.028 = 35.714$). Indeed, Bayes factors of these magnitudes are often conventionally described as very strong evidence in favor of the null hypothesis [28]. On the other hand, anecdotal evidence between the training clusters and all the further wellbeing and recovery variables (i.e., muscle soreness: $BF_{10} = 0.381$; sleep quality: $BF_{10} = 0.959$; and stress: $BF_{10} = 0.475$) was observed. Additionally, cluster RUN revealed higher posterior distribution on sleep duration and lower on fatigue, muscle soreness, sleep quality, and stress compared with the other clusters, whereas cluster LOW showed larger posterior distributions to all wellness variables, except sleep duration.

## Discussion

This study described professional soccer players' training responses during a competitive season to investigate the relationship between these different responses with wellbeing and recovery indices. Results provided insightful information about how training profiles varied during the season, as measured by training load variables. Furthermore, and contrary to what might be expected, findings revealed no substantial connection between training responses with wellbeing and recovery variables. These results may suggest that players' perceived sleep quality and duration, fatigue, muscle soreness, and stress data, are not accounted for when planning and optimizing the training process.

One of the major outcomes of this study was that a data-centered approach allows contemporary soccer demands to be classified into different profiles that contribute to the players' overall development. Accordingly, as evidenced by our training clusters, which exhibited similar and homogeneous profiles in the acceleration variable, as well as in the high level of impacts and body load revealed by the cluster RUN, coaches frequently implement training tasks that mimic the evolving physical nature of the match [3, 4]. The intention is to induce specific adaptations (i.e., improve and optimize soccer players' acceleration capability) that equips players to better cope with match intensity demands, as well as to reduce their vulnerability to injuries [3–5, 7]. Consequently, medical staff and sport researchers stress the importance of

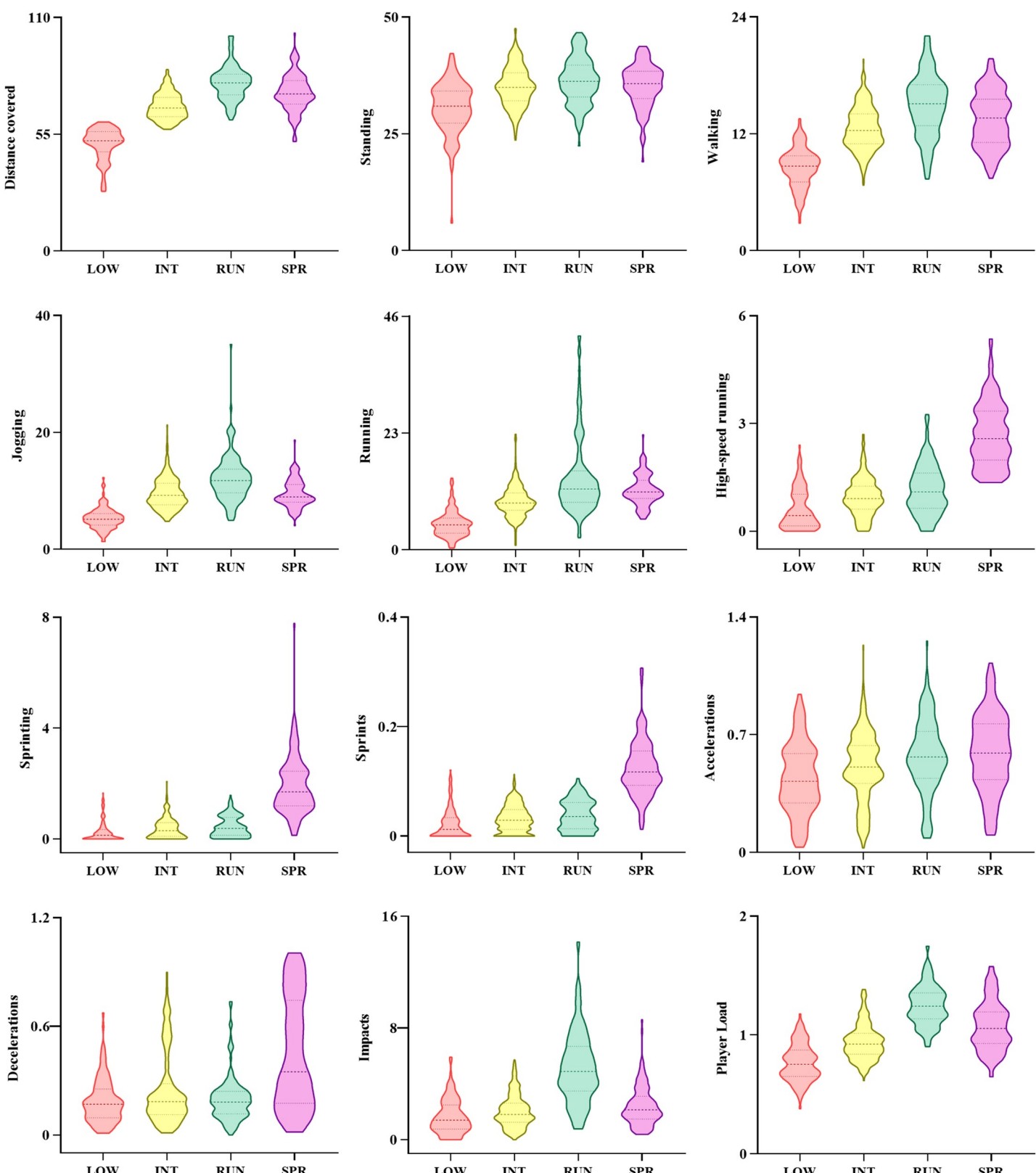

**Fig 3. Violin plots showing the distribution of the training variables according to the clusters obtained.** The violin plots indicate the data distribution, the median and the 1st quartile (25th percentile) and the 3rd quartile (75th percentile), for each group. LOW = lower-demand sessions; INT = intermediate-demand sessions; RUN = running-based sessions; SPR = sprint-based sessions.

**Table 2. Model comparison (cluster distribution) for the Bayesian ANOVA of the influence of the wellbeing and recovery variables.**

| Models | P (M|data) | $BF_M$ | $BF_{10}$ | error % |
|---|---|---|---|---|
| **Sleep duration** | | | | |
| Null model | 0.974 | 37.244 | 1.000 | |
| Training clusters | 0.026 | 0.027 | 0.027 | 0.017 |
| **Fatigue** | | | | |
| Null model | 0.973 | 35.702 | 1.000 | |
| Training clusters | 0.027 | 0.028 | 0.028 | 0.015 |
| **Muscular soreness** | | | | |
| Null model | 0.724 | 2.625 | 1.000 | |
| Training clusters | 0.276 | 0.381 | 0.381 | 6.8E-04 |
| **Sleep quality** | | | | |
| Null model | 0.510 | 1.043 | 1.000 | |
| Training clusters | 0.490 | 0.959 | 0.959 | 3.5E-05 |
| **Stress** | | | | |
| Null model | 0.678 | 2.104 | 1.000 | |
| Training clusters | 0.322 | 0.475 | 0.475 | 4.4E-04 |

The prior model probabilities were all equal (0.5).

Abbreviations: P (M|data) = posterior model probability; $BF_M$ = posterior model odds; $BF_{10}$ = Bayes factor.

short-term high-intensity activities, as an effective training method to minimize injuries [3, 4, 20]. This strategy is corroborated by our data, particularly by the cluster SPR, which is highlighted by high-speed running, sprints and decelerations, confirming that soccer players are frequently exposed to this kind of stimulus, to reduce the likelihood of injury and enhance performance. Nevertheless, frequently challenging the boundaries of what players can achieve, make them more vulnerable to injury, particularly when spikes in loads occur [5, 15, 20], whereby high-intensity stimulus is not sensible throughout periods of tight competitive schedules [9, 10, 13]. Thus, this can be a potential explanation for the cluster INT, which assembles 40% of the training records, suggesting that soccer training load remains constant across the season, with emphasis on technical-tactical improvement and the preservation of the strength and conditioning levels developed during the preseason [6, 12]. The response profiles of cluster LOW illustrated a lower training load, among all others. Considering that training load distribution is heavily regulated by the amount of days preceding a match [6, 10, 11, 30], these results appear to be a coach's strategy to include preventive measures in regular training periodization (e.g., recovering activities, unload the players immediately before and after the match day, etc.), facilitating the reduction of accumulated fatigue, and ultimately promote readiness to perform. Additionally, it is not unreasonable to propose that some discrepancies in our training profiles may illustrate different training load demands among players' health status and playing positions [6, 9–11], as coaches regularly implement training drills that arrange the players according to their condition and match duties.

Factors in addition to training load are likely to inhibit players' performance [17, 18, 21, 22]. In this context, sports science literature suggests that higher sleep quality and proper fitness wellbeing may enhance players' physical and technical performance, while sleep disturbance, muscular soreness, and fatigue accumulation have been associated with performance impairment and higher musculoskeletal injury risk [17, 18, 21, 22]. Nevertheless, this study found an improbable relationship between players' wellness markers and training responses (i.e., despite cluster LOW exposed higher symptoms of muscular soreness, fatigue, and stress,

**Table 3. Descriptive analysis of the model averaged posterior distributions.**

| Variables | Mean ± SD | 95% Credible Interval | |
|---|---|---|---|
| | | Lower | Upper |
| **Sleep duration** | | | |
| Cluster LOW | 7.49±1.11 | 7.30 | 7.69 |
| Cluster INT | 7.38±1.32 | 7.23 | 7.54 |
| Cluster RUN | 7.61±1.30 | 7.40 | 7.82 |
| Cluster SPR | 7.39±1.31 | 7.17 | 7.62 |
| **Fatigue** | | | |
| Cluster LOW | 5.44±1.23 | 5.23 | 5.66 |
| Cluster INT | 5.39±1.10 | 5.27 | 5.53 |
| Cluster RUN | 5.22±1.18 | 5.03 | 5.41 |
| Cluster SPR | 5.29±1.11 | 5.10 | 5.48 |
| **Muscular soreness** | | | |
| Cluster LOW | 5.56±1.28 | 5.33 | 5.78 |
| Cluster INT | 5.43±1.19 | 5.29 | 5.57 |
| Cluster RUN | 5.13±1.30 | 4.92 | 5.34 |
| Cluster SPR | 5.36±1.14 | 5.16 | 5.55 |
| **Sleep quality** | | | |
| Cluster LOW | 5.79±1.11 | 5.59 | 5.99 |
| Cluster INT | 5.69±1.14 | 5.56 | 5.83 |
| Cluster RUN | 5.37±1.19 | 5.17 | 5.56 |
| Cluster SPR | 5.59±1.16 | 5.39 | 5.78 |
| **Stress** | | | |
| Cluster LOW | 5.79±0.99 | 5.62 | 5.98 |
| Cluster INT | 5.70±1.01 | 5.58 | 5.82 |
| Cluster RUN | 5.45±1.05 | 5.28 | 5.62 |
| Cluster SPR | 5.53±1.19 | 5.33 | 5.74 |

which would explain the lower training load), precluding to create player profiles with both training responses and wellness variables. Despite an ever-increasing awareness among coaching staff about the necessity to control players' wellbeing and recovery indices, our findings suggest that professional soccer training on a daily basis is not planned according to the players pre-training condition, therefore training responses are more reliant on the training load that is periodized weekly by the coach than by the players' wellness status. This is concerning, as one day of sleep disturbance and insufficient recovery may not be immediately problematic, but prolonged inadequate sleep habits, accumulated fatigue, and muscle soreness may have severe consequences on players' performance, recovery, and health [31]. Additionally, the inexistence of relation among the different wellbeing and recovery variables (i.e., sleep disturbance could result in insufficient recovery and higher fatigue and muscle pain), was also observed. Therefore, it is not unreasonable to propose that all these findings may indicate reliability issues of self-reported tools or an over/underestimate by the players about their status, suggesting a need to increase the knowledge and experience of both players and coaching staff when implementing and interpret subjective monitoring procedures. Furthermore, the questionnaire being completed 30 minutes before training might not provide enough time for the coach to analyze, interpret and adjust the load, therefore may require rethinking.

Although this study adds relevant findings regarding professional soccer training responses and their relationship with wellness markers, the interpretation of these results should be

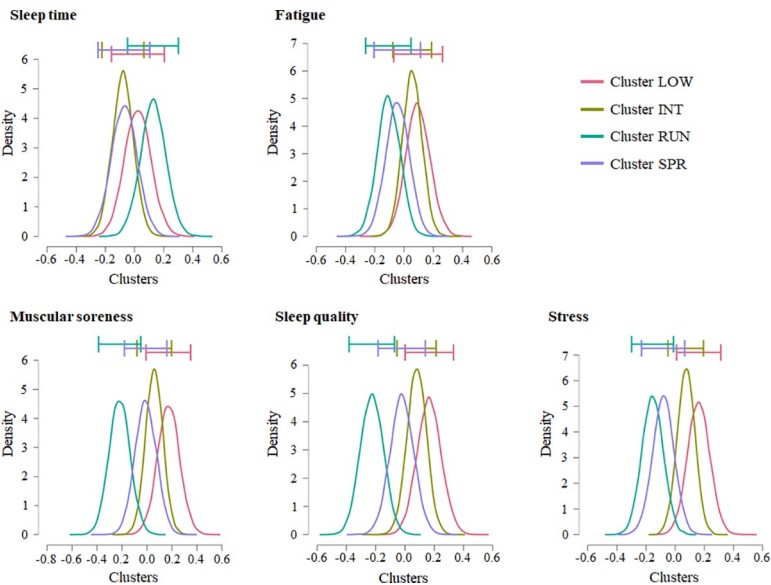

**Fig 4. Model averaged posterior distributions and credible intervals of the relationship between the clusters and the wellbeing and recovery variables.** LOW = lower-demand sessions; INT = intermediate-demand sessions; RUN = running-based sessions; SPR = sprint-based sessions.

taken cautiously, considering that only thirteen players from a single team were involved. A second limitation concerns the non-inclusion of match-day information once the balance of training stimulus has a clear focus on the upcoming match. The non-use of individual speed thresholds is an additional limitation of the current investigation. Furthermore, the use of a single question to assess each wellness variable might be difficult to interpret, whereby an over or underestimation of the players' status should not be ruled out. Therefore, further research is required to better understand players' training responses and the relationship between these different responses with their wellness condition. Accordingly, the use of current technology to measure overall players' wellbeing and health (i.e., physical activity and sedentary behavior during the non-training time) is a promising hot topic [27, 32–34]. This might provide valuable information about players' initial training state and predict the quality of the external output produced during practice, enabling coaches to perform appropriate adjustments if necessary, to balance players' performance and health.

## Conclusions

This study presents new insights into the multifactorial nature of soccer players' training responses throughout the in-season, and the current role of wellness indicators in training planning. The differences in training load highlighted that coaches change training stimuli to improve individual and team performance and to include preventive strategies. Although self-reported methods are attractive and a common tool to identify player's potential pre-training status, our findings suggest an unlikely relationship between the perceived sleep quality and duration, fatigue, muscle soreness, and stress indices, and the training responses profiles. Therefore, the integration of the wellness indicators into the training periodization remains an unclear procedure.

## Author Contributions

**Conceptualization:** Nuno Mateus, Jaime Sampaio.

**Data curation:** Nuno Mateus, Bruno Gonçalves, Javier Sánchez-Sánchez, Jorge Garcia-Unanue.

**Formal analysis:** Nuno Mateus, Bruno Gonçalves, Jose Luis Felipe.

**Methodology:** Nuno Mateus, Jose Luis Felipe, Anthony Weldon, Jaime Sampaio.

**Visualization:** Nuno Mateus.

**Writing – original draft:** Nuno Mateus.

**Writing – review & editing:** Nuno Mateus, Bruno Gonçalves, Jose Luis Felipe, Javier Sánchez-Sánchez, Jorge Garcia-Unanue, Anthony Weldon, Jaime Sampaio.

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
