## [Decision Letter · Decision Letter 0]

13 May 2021

PONE-D-21-01752

In-season training responses and perceived wellbeing and recovery status in professional football players

PLOS ONE

Dear Dr. Mateus,

Thank you for submitting your manuscript to PLOS ONE. After careful consideration, we feel that it has merit but does not fully meet PLOS ONE’s publication criteria as it currently stands. Therefore, we invite you to submit a revised version of the manuscript that addresses the points raised during the review process.

We look forward to receiving your revised manuscript.

Kind regards,

Moacir Marocolo, Ph.D.

Academic Editor

PLOS ONE

Journal Requirements:

Reviewers' comments:

Reviewer's Responses to Questions

**Comments to the Author**

1. Is the manuscript technically sound, and do the data support the conclusions?

Reviewer #1: Yes

Reviewer #2: Yes

Reviewer #3: Partly

2. Has the statistical analysis been performed appropriately and rigorously? 

Reviewer #1: Yes

Reviewer #2: Yes

Reviewer #3: I Don't Know

3. Have the authors made all data underlying the findings in their manuscript fully available?

Reviewer #1: Yes

Reviewer #2: No

Reviewer #3: No

4. Is the manuscript presented in an intelligible fashion and written in standard English?

Reviewer #1: Yes

Reviewer #2: Yes

Reviewer #3: No

5. Review Comments to the Author

Reviewer #1: Dear Authors,

congratulations for your work that shows, besides the benefits of a higher sleep quality in performance the coaches do not take in account during training periodization.

Here are my considerations:

i) Translate all authors afilliations to english

ii) The article is well written

regards

Reviewer #2: The aims of this research study involving 22 male football players were to classify professional football players' training responses during the season and to examine the relationship between these different responses with

wellbeing and recovery indices. No probabilistic interactions were found between training response with wellbeing and recovery markers.

Minor revisions:

1- State and justify the study’s target sample size with a pre-study statistical power calculation.

2- Indicate the date range subject data was collected.

3- Indicate the summary statistics provided in parentheses; i.e., (##.# +/- ##.#). Probably mean and standard deviation.

4- Table 2: Define M|data.

Reviewer #3: Manuscript Number: PONE-D-21-01752

Manuscript Title: In-season training responses and perceived wellbeing and recovery status in professional football players

Overall:

The aim of their study was “to classify professional football players' training responses during the season and to examine the relationship between these different responses with wellbeing and recovery indices”. Although the topic may be relevant, I have several concerns. Not including matches information is a relevant limitation in my opinion, once all preparation is for matches.

I think the rationale (introduction) should be improved dramatically, especially showing why did you do this study? It is not clear on the current form. In the introduction, the link between the premises seems confusing and the aim of the study is not precise. The current aim is inappropriate from a scientific mindset. Researchers classify or examine BECAUSE they have an aim! But per se, “to classify or to examine” are not logical scientific aims.

Maybe, should be better to say: aim was to examine WHETHER the players ‘training responses are associated with wellbeing and recovery indices (?). I am not sure what was/were the aim (s) (?). By the way, the hypothesis is unclear too. If your aim is "only" to describe some variable (s), then okay not having any hypothesis. However, it seems you are willing to make an association between variables?

Specific comments:

The article presents several terms “workload” and you should change for just “load or training load”. Please read Winter E. "Workload"--time to abandon? J Sports Sci. 2006;24(12):1237-8.

Please use “soccer” instead football across the whole manuscript.

You must provide a rationale for your sample size.

Tables and figure MUST be self-exploratory 100% (e.g., the reader not need to go back to the text to understand the abbreviations). I could not find the legends of the figures.

I suggest creating a Experimental Design Figure to make it easier to understand your experiment.

Crucial for the interpretation is the reliability of the measurements. I think that it would strengthen the study if the authors could add ICC an CV values for their most important variables to get an indication of the reliability.

You must provide a rationale for all measurements performed.

Please provide references about validation, reliability for the variables (i.e., daily wellness questionnaire) measured and presented in Figure 1.

More information about the sample must be provided (e.g., VO2max, YOYO performance?). What about the period of the time, number of matches, interval between matches, etc.

The results presentation are confusing. For example, Table 1: distance covered (m) Cluster LOW 50.38±7.55. Does it mean that the players run only ~50 meters? Please make it clearer.

Overall, I can evaluate better the discussion section only if the authors clarify all points that I pointed.

The conclusion is confusing as the whole paper

6. PLOS authors have the option to publish the peer review history of their article (what does this mean?). If published, this will include your full peer review and any attached files.

Reviewer #1: **Yes: **Luis Leitão

Reviewer #2: No

Reviewer #3: No

---

## [Author Response · Author response to Decision Letter 0]

16 Jun 2021

Dear Editor,

Thank you very much for the review of our manuscript entitled: “In-season training responses and perceived wellbeing and recovery status in professional soccer players”. We sincerely appreciate all valuable comments and suggestions, which helped us to improve the quality of the manuscript. Our responses to the Reviewers’ comment are described below in a point-to-point manner. Appropriate changes, suggested by the Reviewers have been introduced to the manuscript (highlighted within the document). 

Yours sincerely,

The Authors

Responses to the reviewer's comments

Reviewer 1

Dear Authors,

Congratulations for your work that shows, besides the benefits of a higher sleep quality in performance the coaches do not take in account during training periodization. 

Here are my considerations:

Translate all authors affiliations to English.

Authors: We understand the reviewer point. The co-author insisted that his university requires that workers' affiliation should be presented in the country's official language. 

The article is well written.

Thank you.

Reviewer 2

The aims of this research study involving 22 male football players were to classify professional football players' training responses during the season and to examine the relationship between these different responses with wellbeing and recovery indices. No probabilistic interactions were found between training response with wellbeing and recovery markers.

Minor revisions:

State and justify the study’s target sample size with a pre-study statistical power calculation.

Authors: We found this as being a relevant consideration since the beginning of the study design. We conducted a Bayesian ANOVA analysis, which does not allow to perform statistical power calculation, neither is so susceptible to the usage of small samples. Nevertheless, we also considered a sample size similar to previous investigations involving professional soccer teams to ensure adequate representativity (Martín-García, A., Díaz, A. G., Bradley, P. S., Morera, F., & Casamichana, D. (2018). Quantification of a professional football team's external load using a microcycle structure. The Journal of Strength & Conditioning Research, 32(12), 3511-3518; Casamichana, D., Martín-García, A., Díaz, A. G., Bradley, P. S., & Castellano, J. Accumulative weekly load in a professional football team: with special reference to match playing time and game position. Biology of Sport, 38(1), 115-124).

Indicate the date range subject data was collected.

Authors: Thank you. The information was added accordingly (please, see P6 L137). The text now reads: “Training load and pre-training wellness indices data for all players were collected over a 16-week in-season period (November-February), without any intervention of the research team regarding training volume and intensity, neither to wellness status perceived by the players (Fig 1).”.

Indicate the summary statistics provided in parentheses; i.e., (##.# +/- ##.#). Probably mean and standard deviation.

Authors: Thank you. The information was added accordingly (please, see P9 L221 – P10 L225). 

Table 2: Define M|data.

Authors: Thank you. The information was added accordingly (please, see Table 2 legend, P13).

Reviewer 3

Manuscript Number: PONE-D-21-01752

Manuscript Title: In-season training responses and perceived wellbeing and recovery status in professional football players

Overall: The aim of their study was “to classify professional football players' training responses during the season and to examine the relationship between these different responses with wellbeing and recovery indices”. Although the topic may be relevant, I have several concerns. Not including matches information is a relevant limitation in my opinion, once all preparation is for matches.

Authors: We agree that matches’ information is important. We tried, but ended up not using it. In most of the cases, football teams do not use the same tracking system in matches and training sessions. In face of this difference, the literature and anecdotal reports from practitioners advised against the risk of analysing data with this instrument bias and, thus, we have not included the data from the matches.

I think the rationale (introduction) should be improved dramatically, especially showing why did you do this study? It is not clear on the current form. In the introduction, the link between the premises seems confusing and the aim of the study is not precise. The current aim is inappropriate from a scientific mindset. Researchers classify or examine BECAUSE they have an aim! But per se, “to classify or to examine” are not logical scientific aims. Maybe, should be better to say: aim was to examine WHETHER the players ‘training responses are associated with wellbeing and recovery indices (?). I am not sure what was/were the aim (s) (?). 

Authors: Thank you. We believe that our message was not clear due to language problems. The information was changed accordingly (please, see P5 L112). The text now reads: “Consequently, the purpose of this study was to describe professional soccer players' training responses during the in-season and to investigate whether these different responses are associated with the players’ wellbeing and recovery indices.”.

By the way, the hypothesis is unclear too. If your aim is "only" to describe some variable (s), then okay not having any hypothesis. However, it seems you are willing to make an association between variables?

Authors: Thank you. The information was changed accordingly (please, see P5 L107). The text now reads: “We hypothesized that coaches’ application of training methods, that mimic game demands and include preventive strategies (e.g., unload players prior to match days) will result in different training loads. Furthermore, proper sleep and wellbeing may be associated with higher training stimulus, while poor sleep and recovery indices may lead to lower training loads as a coaching strategy to cope with the players' pre-training wellness status.”.

Specific comments:

The article presents several terms “workload” and you should change for just “load or training load”. Please read Winter E. "Workload"--time to abandon? J Sports Sci. 2006;24(12):1237-8.

Authors: Relevant consideration. Thank you. The manuscript was changed accordingly. 

Please use “soccer” instead football across the whole manuscript.

Authors: Thank you. The manuscript was changed accordingly. 

You must provide a rationale for your sample size.

Authors: The professional squad used as a sample was constituted of twenty-two outfield players. Football is a contact team sport and, at the highest level of competition, it is very difficult to have the possibility of monitoring the players in long periods of time without disturbance from injuries or medications that might affect their responses. For example, nine of the players did not accomplish the criteria for inclusion or were medicated during the study period or had prolonged injuries. These players had to be excluded from the sample, which leaded to a final sample of thirteen players.

Tables and figure MUST be self-exploratory 100% (e.g., the reader not need to go back to the text to understand the abbreviations). I could not find the legends of the figures.

Authors: Thank you. The information was added accordingly (please, see P12 L233, P13 Table 2, P14 L255). 

I suggest creating a Experimental Design Figure to make it easier to understand your experiment.

Authors: Thank you. The Experimental Design Figure was added accordingly (please, see P7 L152).

Crucial for the interpretation is the reliability of the measurements. I think that it would strengthen the study if the authors could add ICC an CV values for their most important variables to get an indication of the reliability.

Authors: Thank you. The information related to CV was added accordingly (please, see Table 1, P11).

You must provide a rationale for all measurements performed.

Authors: Thank you. In the Data Collection section, the authors reported information about the validation of the variables. Furthermore, Sports Science literature is unanimous in describing the reliability for the variables. Indeed, previous research has applied similar procedures, with the same variables described (Oliveira, R., et al. (2019). In-season training load quantification of one-, two-and three-game week schedules in a top European professional soccer team. Physiology & behavior, 201, 146-156.). 

Please provide references about validation, reliability for the variables (i.e., daily wellness questionnaire) measured and presented in Figure 1.

Authors: Thank you. The information was added accordingly (please, see P8 L180). The text now reads: “The questionnaire was designed to be brief, precise, and based on the components of self-perceived tools used to assess players’ wellness in the literature [23, 24].”.

More information about the sample must be provided (e.g., VO2max, YOYO performance?). What about the period of the time, number of matches, interval between matches, etc.

Authors: We agree with your comment, it is a limitation of the study. The team studied had important competitive goals. It is very difficult to have those goals compatible with ongoing research. Therefore, the coaching staff required the research team to not influence the training sessions (it was impossible to assess players’ VO2 max). Besides, when asked about the possibility of informing the research team about the results of the assessment tests carried out at the beginning of the pre-season, the club did not authorize our request.

Regarding the date range in which the data was collected, information was added accordingly (please, see P6 L137). The text now reads: “Training load and pre-training wellness indices data for all players were collected over a 16-week in-season period (November-February), without any intervention of the research team regarding training volume and intensity, neither to wellness status perceived by the players (Fig 1).”.

The results presentation are confusing. For example, Table 1: distance covered (m) Cluster LOW 50.38±7.55. Does it mean that the players run only ~50 meters? Please make it clearer.

Authors: As described in the Materials and Methods section, all the training load variables were normalized according to the time on the pitch, during each training session to provide an understanding of session intensity (Mateus, N., Exel, J., Gonçalves, B., Weldon, A., & Sampaio, J. (2021). Off-training physical activity and training responses as determinants of sleep quality in young soccer players. Scientific Reports, 11(1), 1-10; Malone S, Owen A, Newton M, Mendes B, Tiernan L, Hughes B, et al. Wellbeing perception and the impact on external training output among elite soccer players. Journal of science and medicine in sport. 2018;21(1):29-34).

---

## [Decision Letter · Decision Letter 1]

22 Jun 2021

PONE-D-21-01752R1

In-season training responses and perceived wellbeing and recovery status in professional soccer players

PLOS ONE

Dear Dr. Mateus,

Thank you for submitting your manuscript to PLOS ONE. After careful consideration, we feel that it has merit but does not fully meet PLOS ONE’s publication criteria as it currently stands. Therefore, we invite you to submit a revised version of the manuscript that addresses the points raised during the review process.

We look forward to receiving your revised manuscript.

Kind regards,

Moacir Marocolo, Ph.D.

Academic Editor

PLOS ONE

Journal Requirements:

Reviewers' comments:

Reviewer's Responses to Questions

**Comments to the Author**

1. If the authors have adequately addressed your comments raised in a previous round of review and you feel that this manuscript is now acceptable for publication, you may indicate that here to bypass the “Comments to the Author” section, enter your conflict of interest statement in the “Confidential to Editor” section, and submit your "Accept" recommendation.

Reviewer #2: All comments have been addressed

Reviewer #3: All comments have been addressed

2. Is the manuscript technically sound, and do the data support the conclusions?

Reviewer #2: (No Response)

Reviewer #3: Partly

3. Has the statistical analysis been performed appropriately and rigorously? 

Reviewer #2: (No Response)

Reviewer #3: N/A

4. Have the authors made all data underlying the findings in their manuscript fully available?

Reviewer #2: (No Response)

Reviewer #3: No

5. Is the manuscript presented in an intelligible fashion and written in standard English?

Reviewer #2: (No Response)

Reviewer #3: Yes

6. Review Comments to the Author

Reviewer #2: (No Response)

Reviewer #3: Dear authors:

Please include the limitations on the end of the discussion section. For example: Not including matches information is a relevant limitation in my opinion, once all preparation is for matches.

More information about the sample must be provided (e.g., VO2max, YOYO

performance?). What about the period of the time, number of matches, interval

between matches, etc.

Authors: We agree with your comment, it is a limitation of the study.

7. PLOS authors have the option to publish the peer review history of their article (what does this mean?). If published, this will include your full peer review and any attached files.

Reviewer #2: No

Reviewer #3: No

---

## [Author Response · Author response to Decision Letter 1]

27 Jun 2021

Dear Editor,

Thank you very much for the review of our manuscript entitled: “In-season training responses and perceived wellbeing and recovery status in professional soccer players”. We sincerely appreciate all valuable comments and suggestions, which helped us to improve the quality of the manuscript. Our responses to the Reviewers’ comment are described below in a point-to-point manner. Appropriate changes, suggested by the Reviewers have been introduced to the manuscript (highlighted within the document). 

Yours sincerely,

The Authors

Responses to the reviewer's comments

Reviewer 3

Reviewer3: Please include the limitations on the end of the discussion section. For example: Not including matches information is a relevant limitation in my opinion, once all preparation is for matches.

Authors: Thank you. The information was added accordingly (please, see P18 L332). The text now reads: “A second limitation concerns the non-inclusion of match-day information once the balance of training stimulus has a clear focus on the upcoming match.”

Reviewer3: More information about the sample must be provided (e.g., VO2max, YOYO performance?). What about the period of the time, number of matches, interval between matches, etc.

Authors: Relevant consideration. However, the team studied had important competitive goals. Hence, it is difficult to have those goals compatible with ongoing research. Therefore, the coaching staff required the research team to not influence the training sessions (it was impossible to assess players’ VO2 max). Besides, when asked about the possibility of informing the research team about the results of the assessment tests carried out at the beginning of the pre-season, the club did not authorize our request.

Regarding the data collection period, the number of matches and the interval between matches in which the data was collected, information was added accordingly (please, see P6 L137). The text now reads: “Training load and pre-training wellness indices data for all players were collected over a 16-week in-season period (November-February), without any intervention of the research team regarding training volume and intensity, neither to wellness status perceived by the players (Fig 1). Participants trained on a full-time basis and played competitive fixtures within the Spanish Second Division and Copa del Rey during the 2017-2018 season. Throughout the data-collection period, the team competed in 17 official matches, which often meant that the team played 1 match per week.”.

---

## [Decision Letter · Decision Letter 2]

1 Jul 2021

In-season training responses and perceived wellbeing and recovery status in professional soccer players

PONE-D-21-01752R2

Dear Dr. Mateus,

We’re pleased to inform you that your manuscript has been judged scientifically suitable for publication and will be formally accepted for publication once it meets all outstanding technical requirements.

Kind regards,

Moacir Marocolo, Ph.D.

Academic Editor

PLOS ONE

Additional Editor Comments (optional):

Reviewers' comments:

Reviewer's Responses to Questions

**Comments to the Author**

1. If the authors have adequately addressed your comments raised in a previous round of review and you feel that this manuscript is now acceptable for publication, you may indicate that here to bypass the “Comments to the Author” section, enter your conflict of interest statement in the “Confidential to Editor” section, and submit your "Accept" recommendation.

Reviewer #3: All comments have been addressed

2. Is the manuscript technically sound, and do the data support the conclusions?

Reviewer #3: Yes

3. Has the statistical analysis been performed appropriately and rigorously? 

Reviewer #3: Yes

4. Have the authors made all data underlying the findings in their manuscript fully available?

Reviewer #3: No

5. Is the manuscript presented in an intelligible fashion and written in standard English?

Reviewer #3: Yes

6. Review Comments to the Author

Reviewer #3: (No Response)

7. PLOS authors have the option to publish the peer review history of their article (what does this mean?). If published, this will include your full peer review and any attached files.

Reviewer #3: No

---

## [Editor Report · Acceptance letter]

6 Jul 2021

PONE-D-21-01752R2 

In-season training responses and perceived wellbeing and recovery status in professional soccer players 

Dear Dr. Mateus:

I'm pleased to inform you that your manuscript has been deemed suitable for publication in PLOS ONE. Congratulations! Your manuscript is now with our production department. 

Kind regards, 

on behalf of

Dr Moacir Marocolo 

Academic Editor

PLOS ONE